# One Policy to Run Them All: an End-to-end Learning Approach to Multi-Embodiment Locomotion

**Nico Bohlinger**[1]**, Grzegorz Czechmanowski**[2,4]**, Maciej Krupka**[2]**, Piotr Kicki**[2,4]
**Krzysztof Walas**[2,4]**, Jan Peters**[1,3,5,6]**, Davide Tateo**[1]
[1] Department of Computer Science, Technical University of Darmstadt, Germany
[2] Institute of Robotics and Machine Intelligence, Poznan University of Technology, Poland
[3] German Research Center for AI (DFKI), Research Department: Systems AI for Robot Learning
[4] IDEAS NCBR, Warsaw, Poland; [5] Hessian.AI; [6]Centre for Cognitive Science

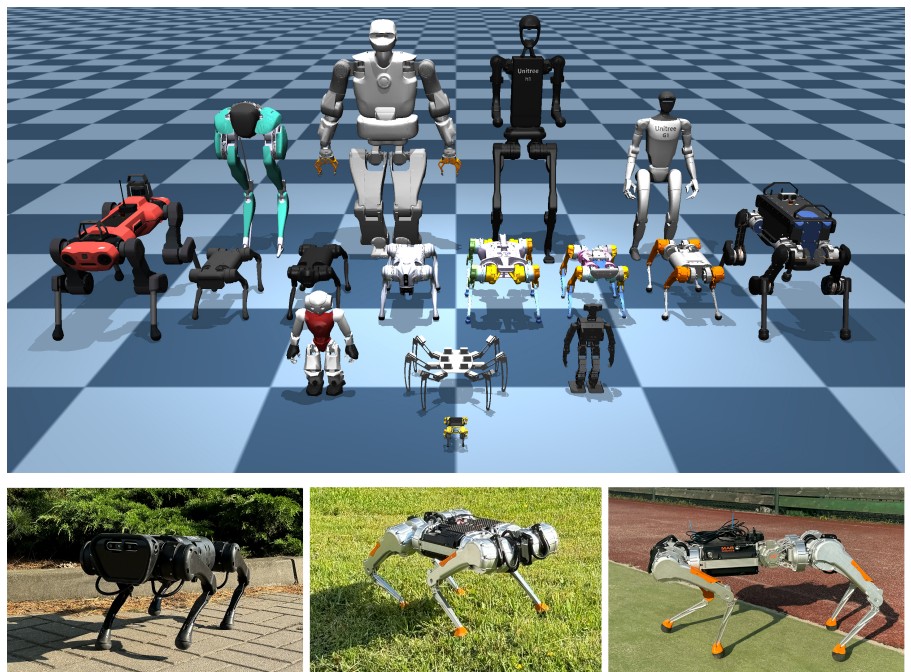

Figure 1: Top – We train a single locomotion policy for multiple robot embodiments in simulation. Bottom – We can transfer and deploy the policy on three real-world platforms by randomizing the embodiments and environment dynamics during training.

**Abstract:** Deep Reinforcement Learning techniques are achieving state-of-the-art results in robust legged locomotion. While there exists a wide variety of legged platforms such as quadruped, humanoids, and hexapods, the field is still missing a single learning framework that can control all these different embodiments easily and effectively and possibly transfer, zero or few-shot, to unseen robot embodiments. We introduce URMA, the Unified Robot Morphology Architecture, to close this gap. Our framework brings the end-to-end Multi-Task Reinforcement Learning approach to the realm of legged robots, enabling the learned policy to control any type of robot morphology. The key idea of our method is to allow the network to learn an abstract locomotion controller that can be seamlessly shared between embodiments thanks to our morphology-agnostic encoders and decoders. This flexible architecture can be seen as a potential first step in building a foundation model for legged robot locomotion. Our experiments show that URMA can learn a locomotion policy on multiple embodiments that can be easily transferred to unseen robot platforms in simulation and the real world.

**Keywords:** Locomotion, Reinforcement Learning, Multi-embodiment Learning

8th Conference on Robot Learning (CoRL 2024), Munich, Germany.

# 1  Introduction

The robotics community has mastered the problem of robust gait generation in the last few years. With the help of Deep Reinforcement Learning (DRL) techniques, legged robots can show impressive locomotion skills. There are numerous examples of highly agile locomotion with quadrupedal robots [1, 2, 3, 4, 5, 6], learning to run at high speeds, jumping over obstacles, walking on rough terrain, performing handstands, and completing parkour courses. Achieving these agile movements is often enabled by training in many parallelized simulation environments and using carefully tuned or automatic curricula on the task difficulty [7, 8]. Even learning simple locomotion behaviors directly on real robots is possible but requires far more efficient learning approaches [9, 10]. Similar methods have been applied to generate robust walking gaits for bipedal and humanoid robots [11, 12, 13]. The learned policies can be effectively transferred to the real world and work in all kinds of terrain with the help of extensive Domain Randomization (DR) [14, 15] during training. Additionally, techniques like student-teacher learning [1, 16] or the addition of model-based components [17, 18] or constrains [19, 20, 21] to the learning process can further improve the learning efficiency and robustness of the policies.

At the same time, new advances in computational power, the availability of large datasets, and the development of foundation models are opening new frontiers for artificial intelligence, allowing us to implement and learn more complex and intelligent agent behaviors. Future robots will require incorporating these models into the control pipeline [22, 23]. However, to fully benefit from foundation models, we need to be able to integrate these high-level policies with the low-level control of the robots. The long-term objective would be to develop foundation models for locomotion, allowing zero-shot (or few-shot) deployment to any arbitrary platform. However, to reach this objective, it is fundamental to adapt the underlying learning system to support different tasks and morphologies. Therefore, we argue that the Multi-Task Reinforcement Learning (MTRL) problem is a fundamental topic for the future research of robot locomotion, and, indeed, this formulation has recently attracted the interest of the community, using both structured [24] and end-to-end learning approaches [25]. MTRL algorithms share knowledge between tasks and learn a common representation space that can be used to solve all of them [26, 27]. To map differently sized observation and action spaces into and out of the shared representation space, implementations often resort to padding the observations and actions with zeros to fit a maximum length [28] or to using a separate neural network head for each task [26]. These methods allow for efficient training but can be limiting when trying to transfer to new tasks or environments: for every new robot, the training process has to be repeated from scratch, as different embodiments require different hyperparameters, reward coefficients, training curricula, etc. Already in the case of the same robot morphology, e.g. quadrupeds, a trained policy can not be easily transferred when the number of joints is not the same for the robots. This is even more evident when trying to reuse the learned gait across different types of morphologies. This issue is closely related to the fundamental correspondence problem in robotics [29], as the policy has to learn an internal mapping between the different action and observation spaces and the embodiments themselves, which define the robots' kinematics. In practice, the number of joints and feet of a legged robot determines the size of its action and observation space, which can differ for every new robot. This often prevents a straightforward transfer of existing policies as the learning architecture fully depends on the specific robot platform.

To tackle this problem and to move to more powerful and general policies that can be used as locomotion foundation models, we propose a novel MTRL framework that allows simultaneously learning locomotion tasks with many different morphologies easily and effectively. Our approach is based on a novel neural network architecture that can handle differently sized action and observation spaces, allowing the policy to adapt easily to diverse robot morphologies. Furthermore, our method allows us zero-shot deployment of the policy to unseen robots and few-shot fine-tuning on novel target platforms. We highlight the effectiveness of our approach, first with a theoretical analysis and then by training a single locomotion policy on 16 robots, including quadrupeds, hexapods, bipeds, and humanoids. Finally, we zero-shot transfer the learned policy to two simulated and three real-world robots, showing the transferability and robustness of our method.

**Related Work**

Early work on controlling different robot morphologies is based on the idea of using Graph Neural Networks (GNNs) to capture the morphological structure of the robots [30, 31, 32]. Each node in the graph represents a joint of a robot, and its state is comprised of the joint's specific information, e.g. current position, velocity, etc. Through message passing, the GNN can then aggregate information from neighboring joints and learn to control the robot as a whole. GNN-based approaches can control different robots even when removing some of their limbs, but they struggle to generalize to many different morphologies at once, as every morphology requires a different graph structure. Furthermore, the local nature of message passing can lead to information bottlenecks in the policies and the inability to act as a cohesive global controller [33].

Transformer-based architectures have been proposed to overcome the limitations of GNNs by using the attention mechanism to globally aggregate information of varying numbers of joints [33, 25, 34, 35]. These methods still lack substantial generality as they are limited to morphologies that were defined a priori. For example, Kurin et al. [33] uses encoder-decoder pairs for each type of joint, which limits the architecture to a set of predefined joint types and does not allow for components that only have associated observations but no actions, e.g. the feet of legged robots. Trabucco et al. [25] defines tokens for each type of observation and joint, which is a more general system, but those tokens are handcrafted, and there is a different set of joint tokens for every morphology, which makes generalization between morphologies and to new ones difficult or even impossible.

Besides the GNN- and Transformer-based methods, which consider only environments with 2D-planar or physically implausible robots, Shafiee et al. [24] recently showed that a single controller can be trained to control 16 different 3D-simulated quadrupedal robots and to transfer to two of them in the real world. Their method uses a Central Pattern Generator (CPG) and inverse kinematics to generate and track trajectories of the four feet of the robots. This approach has the drawback of discarding the joint-specific information, and the controller can be deployed only on robots with the same number of feet. Furthermore, Feng et al. [36] and Luo et al. [37] use procedurally generated quadrupeds in simulation to train a single policy that transfers to unseen real-world quadrupeds. However, their learning framework assumes that every robot is in the same morphology class, and their neural network architecture can only deal with robots with the same number of joints. Compared to all the other approaches, our method can handle multiple embodiments from any legged morphology, can adapt to arbitrary joint configurations with the same network and can be deployed on real world robots.

## 2 Multi-embodiment Locomotion with a Single Policy

In MTRL the objective is to learn a single policy $\pi_\theta$ that optimizes the average of the expected discounted return $\mathcal{J}^m(\boldsymbol{\theta})$ over the reward function $r^m$ across $M$ tasks:

$$\mathcal{J}(\boldsymbol{\theta}) = \frac{1}{M} \sum_m^M \mathcal{J}^m(\boldsymbol{\theta}), \qquad \mathcal{J}^m(\boldsymbol{\theta}) = \mathop{\mathbb{E}}_{\tau \sim \pi} \left[ \sum_{t=0}^T \gamma^t r^m(s, a) \right], \qquad (1)$$

where $\tau$ is a trajectory given by the state-action pairs $(s_t, a_t)$, $\gamma$ is the discount factor, and $T$ is the time horizon. In our case, we consider different robot embodiments as separate tasks and train a policy controlling all robots and optimizing the objective described in (1). We aim to design a policy where the underlying neural network architecture is independent of the set of possible embodiments. Therefore, we propose the Unified Robot Morphology Architecture (URMA), which is completely morphology agnostic, i.e. it can be applied to any type of robot with any number of joints such that there is no need to define the possible morphologies or joints beforehand. We use URMA to learn robust locomotion policies, but its formulation is general enough to be applied to any control task. Figure 2 presents a schematic overview of URMA. In general, URMA splits the observations of a robot into distinct parts, encodes them with a simple attention encoder [38] with a learnable temperature [39], and uses our universal morphology decoder to obtain the actions for every joint of the robot.

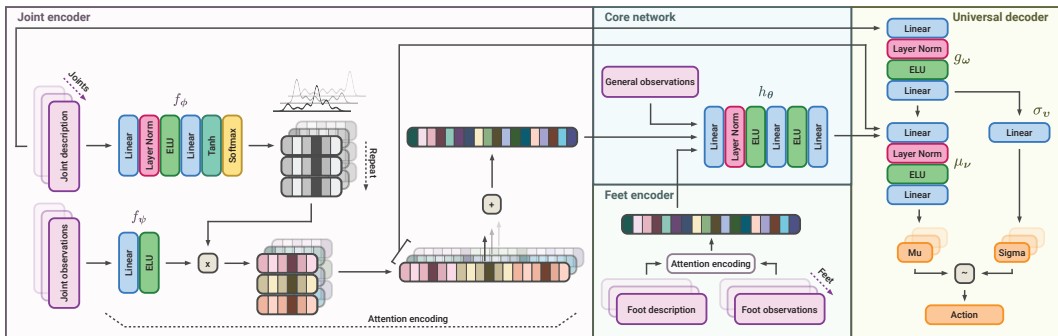

Figure 2: Overview of URMA. Left – Joint observations and descriptions are encoded and combined into a single joint latent vector through an attention head. Bottom center – Feet observations and descriptions are encoded in the same way. Top center – Joint latent, feet latent, and general observations are fed through the core network to get the action latent vector. Right – The universal morphology decoder encodes the joint descriptions and pairs them with the action latent vector and the single joint latent vector to produce the action mean and standard deviation for the final action.

To handle observations of any morphology, URMA first splits the observation vector $o$ into robot-specific and general observations $o_g$, where the former can be of varying size, and the latter has a fixed dimensionality. For locomotion, we subdivide the robot-specific observations into joint and feet-specific observations. This split is not necessary but makes the application to locomotion cleaner. In the following text, we describe everything w.r.t. the joint-specific observations, but the same applies to the feet-specific ones as well. Every joint of a robot is composed of joint-specific observations $o_j$ and a description vector $d_j$, which is a fixed-size vector that can uniquely describe the joint by using characteristic properties like the joint's rotation axis, its relative position in the robot, torque and velocity limits, control range, etc. (see Appendix A for more details). The description vectors and joint-specific observations are encoded separately by the Multilayer Perceptrons (MLPs) $f_\phi$ and $f_\psi$ and are then passed through a simple attention head, with a learnable temperature $\tau$ and a minimum temperature $\epsilon$, to get a single latent vector

$$\bar{z}_{\text{joints}} = \sum_{j \in J} z_j, \qquad z_j = \frac{\exp\left(\frac{f_\phi(d_j)}{\tau + \epsilon}\right)}{\sum_{j \in J} \exp\left(\frac{f_\phi(d_j)}{\tau + \epsilon}\right)} f_\psi(o_j), \tag{2}$$

that contains the information of the joint-specific observations of all joints. With the help of the attention mechanism, the network can learn to separate the relevant joint information and precisely route it into the specific dimensions of the latent vector by reducing the temperature $\tau$ of the softmax close to zero. The joint latent vector $\bar{z}_{\text{joints}}$ is then concatenated with the feet latent vector $\bar{z}_{\text{feet}}$ and the general observations $o_g$ and passed to the policies core MLP $h_\theta$ to get the action latent vector $\bar{z}_{\text{action}} = h_\theta(o_g, \bar{z}_{\text{joints}}, \bar{z}_{\text{feet}})$. To obtain the final action for the robot, we use our universal morphology decoder, which takes the general action latent vector and pairs it with the set of encoded specific joint descriptions and the single joint latent vectors to produce the mean and standard deviation of the actions for every joint, from which the final action is sampled as

$$a^j \sim \mathcal{N}(\mu_\nu(d_j^a, \bar{z}_{\text{action}}, z_j), \sigma_\upsilon(d_j^a)), \qquad\qquad d_j^a = g_\omega(d_j). \tag{3}$$

To ensure that only fully normalized and well-behaved observations come into the network, we use LayerNorm [40] after every input layer. The learning process also benefits from adding another LayerNorm in the action mean network $\mu_\nu$. We argue that this choice improves the alignment of the different latent vectors entering into $\mu_\nu$ better. To ensure a fair comparison, we also use LayerNorms with the same rationale in the baseline architectures. We highlight the benefits in the ablation study on the usage of LayerNorm for URMA in Figure 4 of Appendix D.

Our second contribution is the open-source modular learning framework, which enables us to easily train robust and transferable locomotion policies for all kinds of legged robots. When adding a new robot to the training set, only the reward coefficients, controller gains, and domain randomization

ranges have to be adjusted, which can be easily done by slightly modifying the ones from existing robots in the framework. As the penalty terms in the reward function are not essential for learning the core locomotion but only shape the resulting gait, we apply a time-dependent fixed-length curriculum $r_c(t) = \min\left(\frac{t}{T}, 1\right) r_c^T$, where $t$ is the current training step, $T$ is the curriculum length, and $r_c^T$ is the final penalty coefficient. This speeds up the learning and makes the coefficient tuning process easier and more forgiving, as the policy can handle higher penalties better when it has already learned to perform basic locomotion. The full details on the environment setup, reward function, and domain randomization can be found in Appendix A.

**Theoretical Analysis**

To evaluate the benefit of learning shared representations across robot morphologies through URMA, we extend the multi-task risk bounds from Maurer et al. [41] and D'Eramo et al. [26] to our morphology-agnostic encoder and decoder. As we will use the Proximal Policy Optimization (PPO) algorithm in our learning framework, we frame our problem as an evaluation of the performance difference between the PPO training on the empirical dataset and the optimal policy update with infinite samples. In our simplified analysis, we will assume that i. our policy optimization step can find the policy that minimizes the surrogate loss over the current dataset; ii. the trust region is small enough such that the surrogate loss and the expected discounted return of the policy are close enough; iii. the surrogate loss is computed with the true advantage. Given our set of robot embodiments $\boldsymbol{\mu} = (\mu_1, \ldots, \mu_M)$, the set $\bar{\mathbf{X}} \in \mathcal{X}^{Mn}$ of $n$ input samples from the space of observations, descriptions and actions $\mathcal{X}$ for each of the $M$ tasks and the corresponding set $\bar{\mathbf{Y}} \in \mathbb{R}^{Mn}$ of advantages $A^\pi$ divided by the initial probabilities $\pi_0$, we can describe the policy learning as finding the minimizer $\hat{f}, \hat{h}, \hat{w}$ of the following optimization problem:

$$\langle \hat{f}, \hat{h}, \hat{w} \rangle = \min_{\langle f, h, w \rangle} \left\{ \frac{1}{nM} \sum_{m=1}^{M} \sum_{i=1}^{n} \ell(w(h(f(X_{mi}))), Y_{mi}) : f \in \mathcal{F}, h \in \mathcal{H}, w \in \mathcal{W} \right\}, \qquad (4)$$

with $f \in \mathcal{F} : \mathcal{X} \to \mathbb{R}^J \times \mathcal{X}$, $h \in \mathcal{H} : \mathbb{R}^J \times \mathcal{X} \to \mathbb{R}^K \times \mathcal{X}$ and $w \in \mathcal{W} : \mathbb{R}^K \times \mathcal{X} \to \mathbb{R}$ being the encoder, core and decoder networks, and $\ell : \mathbb{R} \times \mathbb{R} \to [0, 1]$ the normalized policy optimization loss function. We quantify the performance of the functions $f, h, w$ with the task-averaged risk

$$\varepsilon_{\text{avg}}(f, h, w) = \frac{1}{M} \sum_{m=1}^{M} \mathbb{E}_{(X,Y) \sim \mu_t} \left[ \ell(w(h(f(X))), Y) \right]. \qquad (5)$$

We define $\varepsilon_{\text{avg}}^*$ as the minimum of this risk, with the minimizers $f^*$, $h^*$ and $w^*$. We measure the complexity of some function class $\mathcal{Z}$ composed of $K$ functions via the set $\mathcal{Z}(\bar{\mathbf{X}}) = \{(z_k(X_{mi})) : z \in \mathcal{Z}\} \subseteq \mathbb{R}^{KMn}$ with the Gaussian complexity [42]

$$G(\mathcal{Z}(\bar{\mathbf{X}})) = \mathbb{E}\left[ \sup_{z \in \mathcal{Z}} \sum_{mki} \gamma_{mki} z_k(X_{mi}) | X_{mi} \right], \qquad (6)$$

where $\gamma_{mki}$ are i.i.d. standard Gaussian random variables.

Furthermore, we define $L(\mathcal{Z})$, as the upper bound of the Lipschitz constant of all $z$ in $\mathcal{Z}$, and the Gaussian average of Lipschitz quotients

$$O(\mathcal{Z}) = \sup_{y, y' \in Y, y \neq y'} \mathbb{E}\left[ \sup_{z \in Z} \frac{\langle \gamma, z(y) - z(y') \rangle}{||y - y'||} \right] \qquad (7)$$

where $\gamma$ is a vector of $d$ i.i.d. standard Gaussian random variables and $z \in \mathcal{Z} : Y \to \mathbb{R}^d$ with $Y \subseteq \mathbb{R}^p$. Using the definitions above, we can bound the risk $\varepsilon_{\text{avg}}$:

**Theorem 1.** *Let $\boldsymbol{\mu}$, $\mathcal{F}$, $\mathcal{H}$ and $\mathcal{W}$ be defined as above and assume $0 \in \mathcal{H}$ and $w(0) = 0, \forall w \in \mathcal{W}$. Then for $\delta > 0$ with probability at least $1 - \delta$ in the draw of $(\bar{\mathbf{X}}, \bar{\mathbf{Y}}) \sim \prod_{m=1}^{M} \mu_m^n$ we have that*

$$\varepsilon_{avg}(\hat{f}, \hat{h}, \hat{w}) \leq c_1 \frac{L(\ell)L(\mathcal{W})L(\mathcal{H})G(\mathcal{F}(\bar{\mathbf{X}}))}{nM} + c_2 \frac{L(\ell)L(\mathcal{W}) \sup_f ||f(\bar{\mathbf{X}})||O(\mathcal{H})}{nM}$$

$$+ c_3 \frac{L(\ell)L(\mathcal{W}) \min_{p \in P} G(\mathcal{H}(p))}{nM} + c_4 \frac{L(\ell) \sup_{h,f} ||h(f(\bar{\mathbf{X}}))||O(\mathcal{W})}{nM} + \sqrt{\frac{8 \ln(\frac{3}{\delta})}{nM}} + \varepsilon_{avg}^*. \qquad (8)$$

The proof of Theorem 1 and a discussion on the assumptions can be found in Appendix M. For reasonable function classes $\mathcal{W}$, the Gaussian average of Lipschitz quotients $O(\mathcal{W})$ can be bounded independently from the number of samples. For most settings, the Gaussian complexity $G(\mathcal{F}(\bar{\mathbf{X}}))$ is $\mathcal{O}(\sqrt{nM})$. Also the terms $\sup_f \|f(\bar{\mathbf{X}})\|$ and $\sup_{h,f} \|h(f(\bar{\mathbf{X}}))\|$ are $\mathcal{O}(\sqrt{nM})$, if they can be uniformly bounded. Using these assumptions, the URMA policy structure is better suited for multi-task learning as all the first four terms are $\mathcal{O}\left(1/\sqrt{nM}\right)$. In comparison, the multi-head architecture from from D'Eramo et al. [26] requires additional encoder and decoder heads for every task, and thus, the cost of learning all the encoders and decoders is only $\mathcal{O}\left(1/\sqrt{n}\right)$, i.e. it is not reduced when increasing the number of tasks $M$, as it is in our case for URMA. In conclusion, as URMA only uses a single general encoder and decoder for all tasks, it compares favorably to the typical multi-head approach as it can focus on learning only a single mapping to the shared representation space compared to the multi-head architecture which needs to learn $M$ different encodings and decodings. This leads to the lower sample cost of learning these shared representations.

## 3 Experiments

In this section, we evaluate our method from three different perspectives. First, we assess how well the model learns to control multiple embodiments in parallel against classical MTRL baselines. Then, we analyze how well the URMA architecture performs in terms of zero and few-shot transfer. Finally, we test the deployment capabilities of our learning framework and control architecture on real robots, allowing us to bridge the sim-to-real gap effectively. For all experiments in simulation, we will use the following two baselines.

– **Multi-Head Baseline.** One way of dealing with the issue of different action and observation space sizes is to use a multi-head architecture with an encoder head for every environment, a shared core, and a decoder head for every environment [26]. We implement this baseline by using one shared encoder and decoder for all quadruped robots, one for all humanoid and bipedal robots, and one for the hexapod. The observations of all robots with identical morphology are arranged in the same order for the encoder, and the observations for missing joints are simply set to zero, e.g. humanoids often have different joint configurations. Compared to URMA, this baseline allows for efficient learning as the morphology-specific heads inherently separate the observations and arrange them in the correct order from the beginning. However, introducing new joints or completely new morphologies requires adding new neurons to a head or training a completely new head from scratch. Every additional head is a new mapping into and out of the shared representation space, which leads to a higher learning complexity compared to URMA.

– **Padding Baseline.** Another way to handle differently sized action and observation spaces is to pad the observations and actions with zeros to fit a maximum length [28]. Therefore, a specific observation dimension can now represent different things for different robots. We add a one-hot task ID to the observations to ensure that the policy can distinguish between the robots, as typically done in MTRL [27]. Compared to URMA, the padding baseline is less complex but has similar issues when transferring to new morphologies like the multi-head baseline, as a new robot essentially represents a completely new task, and the policy has a hard time transferring knowledge between the differently structured observations and actions.

To train our locomotion policy, we use the CPU-based MuJoCo physics simulator [43] for 16 different simulated robots with three learning environments each, resulting in 48 parallel environments in total. Figure 1 shows all the simulated robots we utilize for training and the three real robots that we use for deployment in the real world. The set of robots includes nine quadrupeds with three different joint configurations, five humanoids with five different joint configurations, one biped, and one hexapod. To leverage the huge amounts of data that we can generate in simulation, we use the PPO algorithm [44]. We build on the codebase of the DRL library RL-X [45] to implement our architecture and the baselines in JAX and to run the experiments. All hyperparameters used for URMA and the baselines are noted in Table 4 in Appendix B.

**Results**

First, we want to evaluate the training efficiency of MTRL in our setting. We train URMA and the baselines on all robots simultaneously and compare the average return to the single-robot training setting, where a separate policy is trained for every robot. All policies are trained on 100 million steps per robot, and every experiment shows the average return over 5 seeds and the corresponding 95% confidence interval. Additionally, we plot the empirical maximum performance when continuing the training for 1.6B steps on only a single robot as a dashed line. The performance on all 16 robots individually can be found in Figure 13 in Appendix L. Figure 3 confirms the advantage in learning efficiency of MTRL over single-task learning, as URMA and the multi-head baseline learn significantly faster than the average over training only on a single robot at a time. As expected, early on in training, URMA learns slightly slower than the multi-head baseline due to the time needed by the attention layers to learn to separate the robot-specific information, which the multi-head baseline inherently does from the beginning. However, URMA ultimately reaches a higher final performance. The padding baseline performs noticeably worse than the other two. We argue that the policy has trouble learning the strong separation in representation space between the different robots—which is necessary for the differently structured observation and action spaces—only based on the task ID.

Next, we evaluate the zero-shot and few-shot transfer capabilities of URMA and the baselines on two robots that were withheld from the training set of the respective policies. We test the zero-shot transfer on the Unitree A1, a robot whose embodiment is similar to other quadrupeds in the training set. Figure 3 shows the evaluation for the A1 during a training process with the other 15 robots and highlights that both URMA and the multi-head baseline can transfer perfectly well to the A1 while never having seen it during training. The policy trained only on the A1 (shown in black) performs distinctly worse during the 100M training steps as it needs more samples per robot to learn the task. It eventually catches up to the multi-embodiment zero-shot performance while training for 1.6B steps directly on the A1. Further ablations on the minimal set of quadrupeds needed for good zero-shot capabilities on the A1 can be found in Appendix H.

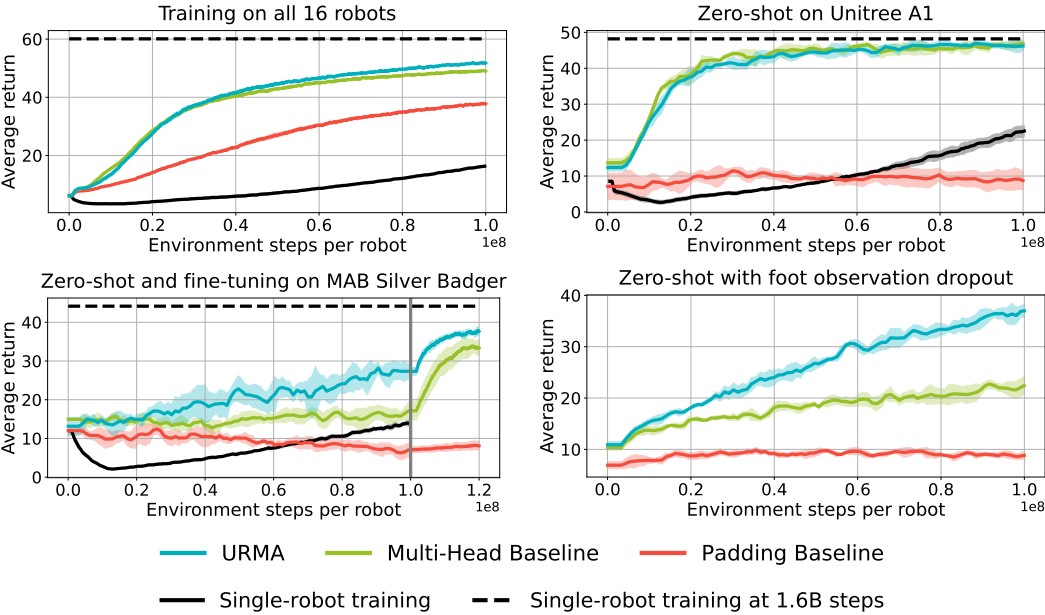

Figure 3: Top left – Average return of the three architectures during training on all 16 robots compared to the single-robot training setting. Top right – Zero-shot transfer to the Unitree A1 while training on the other 15 robots. Bottom left – Zero-shot transfer to the MAB Robotics Silver Badger while training on the other 15 robots and fine-tuning on only the Silver Badger afterward. Bottom right – Zero-shot evaluation on all 16 robots while removing the feet observations.

To investigate an out-of-distribution embodiment, we use the same setup as for the A1 and evaluate zero-shot on the MAB Robotics Silver Badger robot, which has an additional spine joint in the trunk and lacks feet observations, and then fine-tune the policies for 20 million steps only on the Silver Badger itself. Figure 3 shows that URMA can handle the additional joint and the missing feet observations better than the baselines and is the only method capable of achieving a good gait at the end of training. After starting the fine-tuning (gray vertical line), URMA maintains the lead in the average return due to the better initial zero-shot performance. Similar zero-shot and fine-tuning experiments for the humanoid morphology can be found in Appendix I.

To further assess the adaptability of our approach, we evaluate the zero-shot performance in the setting where observations are dropped out, which can easily happen in real-world scenarios due to sensor failures. To test the additional robustness in this setting, we train the architectures on all robots with all observations and evaluate them on all robots while completely dropping the feet observations. Figure 3 confirms the results from the previous experiment with the Silver Badger and shows that URMA can handle missing observations better than the baselines.

Finally, we deploy the same URMA policy on the real Unitree A1, MAB Honey Badger, and MAB Silver Badger quadruped robots. Figure 1 and the videos on the project page show the robots walking with the learned policy on pavement, grass, and plastic turf terrain with slight inclinations. Due to the extensive DR during training, the single URMA policy trained on 16 robots in simulation can be zero-shot transferred to the three real robots without any further fine-tuning. While the Unitree A1 and the MAB Silver Badger are in the training set, the network is not trained on the MAB Honey Badger. Despite the Honey Badger's gait not being as good as the other two robots, it can still locomote robustly on the terrain we tested, proving the generalization capabilities of our architecture and training scheme.

**Limitations.** While our method is the first end-to-end approach for learning multi-embodiment locomotion, many open challenges remain. On one side, our generalization capabilities rely mostly on the availability of data, therefore zero-shot transfer to embodiments that are completely out of the training distribution is still problematic. This issue could be tackled by exploiting other techniques in the literature, such as data augmentation and unsupervised representation learning, to improve our method's generalization capabilities. Furthermore, we currently omit exteroceptive sensors from the observations, which can be crucial to learning policies that can navigate in complex environments and fully exploit the agile locomotion capabilities of legged robots. Lastly, as humanoid robots only recently started to be available for reasonable prices, we could not test the deployment on one of these platforms yet. Better modelling, more reward engineering or additional randomization might be necessary to ensure their real-world transfer.

## 4 Conclusion

We presented URMA, a open-source framework[1] to learn robust locomotion for different types of robot morphologies end-to-end with a single neural network architecture. Our flexible learning framework and the efficient encoders and decoders allow URMA to learn a single control policy for 16 different embodiments from three different legged robot morphologies. We highlight URMAs learning efficiency in a theoretical analysis of its task-averaged risk bound and compare it to prior work. In practice, URMA reaches higher final performance on the training with all robots, shows higher robustness to observation dropout, and better zero-shot capabilities to new robots compared to MTRL baselines. Furthermore, we deploy the same policy zero-shot on two known and one unseen quadruped robot in the real world. We argue that this multi-embodiment learning setting can be easily extended to more complex scenarios and can serve as a basis for locomotion foundation models that can act on the lowest level of robot control. Finally, the URMA architecture is general enough to be applied to not only any robot embodiment but also any control task, making task generalization, also for non-locomotion tasks, an interesting avenue for future research.

---

[1] Project page: https://nico-bohlinger.github.io/one_policy_to_run_them_all_website/

**Acknowledgments**

This project was funded by National Science Centre, Poland under the OPUS call in the Weave programme UMO-2021/43/I/ST6/02711, and by the German Science Foundation (DFG) under grant number PE 2315/17-1. Part of the calculations were conducted on the Lichtenberg high performance computer at TU Darmstadt.

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

# Appendix

## A  Characteristics of the Learning Environment and Robot Embodiments

**Environment Details.**  Every robot is controlled with a PD controller at 50Hz, where the target joint velocity is set to 0, and the target joint position is calculated by: $q^{\text{target}} = q^{\text{nominal}} + \sigma_a a$, with $q^{\text{nominal}}$ being a nominal joint position, $a$ the action of the policy and $\sigma_a$ a scaling factor. We inherit this action scheme from the implementation details from the DRL locomotion literature [2, 7, 16]. The gains $k_p$ and $k_d$ and the scaling factor $\sigma_a$ vary from robot to robot and are noted in Table 1. The observation space is divided into joint-specific, feet-specific, and general observations. The joint-specific observations $o_j$ consist of the joint position $q$, joint velocity $\dot{q}$, and previous action $a_{t-1}$ for each joint. The feet observations $o_f$ include the contact flag $p_f$ and the time since each foot's last contact $p_f^T$. The general observations $o_g$ are the linear velocity of the trunk/torso $v$ (only for the critic), angular velocity $\omega$, command x, y and yaw velocities $c$ (re-sampled twice per episode on average), orientation of the gravity vector $g$, current height above ground $h$ (only for the critic), PD controller variables $k_p, k_d$ and $\sigma_a$, total mass of the robot $m$ and its dimensions: length $L$, width $W$ and height $H$. All observations are concatenated to $o = [o_j, o_f, o_g]$ and normalized to $[-1, 1]$. For URMA, the joint descriptions $d_j$ consist of a joint-specific part, which is made of the relative 3D position in the nominal joint configuration, rotation axis, number of direct child joints, joint nominal position, torque limit, velocity limit, damping, rotor inertia, stiffness, friction coefficient and min and max control range, and a general robot part, which is made of $k_p, k_d, \sigma_a, m, L, W, H$. Similarly, the feet descriptions $d_f$ consist of the relative 3D position of the foot in the nominal joint configuration and of the general robot attributes $k_p, k_d, \sigma_a, m, L, W, H$.

| Morphology | Robot name | J | $k_p$ | $k_d$ | $\sigma_a$ | $m$ | $L$ | $W$ | $H$ |
|---|---|---|---|---|---|---|---|---|---|
| Quadruped | ANYbotics ANYmal B | 12 | 80.0 | 2.0 | 0.5 | 33.3 | 1.21 | 0.81 | 1.37 |
| | ANYbotics ANYmal C | 12 | 80.0 | 2.0 | 0.5 | 45.0 | 1.09 | 0.88 | 0.91 |
| | Google Barkour v0 | 12 | 20.0 | 0.5 | 0.25 | 11.5 | 0.70 | 0.38 | 0.45 |
| | Google Barkour vB | 12 | 30.0 | 0.5 | 0.25 | 11.5 | 0.73 | 0.48 | 0.49 |
| | MAB Silver Badger | 13 | 20.0 | 0.5 | 0.25 | 13.12 | 0.78 | 0.41 | 0.52 |
| | Petoi Bittle | 8 | 25.0 | 0.5 | 0.6 | 0.2 | 0.17 | 0.12 | 0.12 |
| | Unitree A1 | 12 | 20.0 | 0.5 | 0.25 | 12.5 | 0.67 | 0.43 | 0.48 |
| | Unitree Go1 | 12 | 20.0 | 0.5 | 0.25 | 12.7 | 0.70 | 0.42 | 0.48 |
| | Unitree Go2 | 12 | 20.0 | 0.5 | 0.3 | 15.2 | 0.75 | 0.44 | 0.48 |
| Biped | Agility Robotics Cassie | 10 | 70.0 | 2.0 | 0.6 | 33.3 | 0.60 | 0.60 | 1.26 |
| Humanoid | PAL Robotics Talos | 24 | 80.0 | 2.0 | 0.75 | 93.3 | 0.46 | 1.10 | 1.65 |
| | Robotis OP3 | 20 | 21.0 | 0.5 | 0.6 | 3.1 | 0.24 | 0.28 | 0.53 |
| | SoftBank Nao V5 | 22 | 30.0 | 0.5 | 0.6 | 5.3 | 0.17 | 0.43 | 0.59 |
| | Unitree G1 | 23 | 45.0 | 1.0 | 0.5 | 32.2 | 0.29 | 0.55 | 1.26 |
| | Unitree H1 | 19 | 60.0 | 2.0 | 0.75 | 51.4 | 0.55 | 0.83 | 1.77 |
| Hexapod | Custom Hexapod | 18 | 30.0 | 0.5 | 0.6 | 1.9 | 0.43 | 0.56 | 0.24 |

Table 1: Morphology type, robot name, number of controlled joints (J), PD gains, action scaling factor, total mass in kg, length in meter, width in meter and height in meter for every robot in the training set.

**Reward Function.**  The reward function consists of two tracking reward terms for following the command velocities $c$ and multiple penalty terms to shape the gait. All the terms are scaled with individual coefficients, summed up, and clipped to be above zero. Table 2 and Table 3 note the equations for each term and the respective coefficients.

| | Term | Equation |
|---|---|---|
| (T1) | Xy velocity tracking | $\exp(-|v_{xy} - c_{xy}|^2/0.25)$ |
| (T2) | Yaw velocity tracking | $\exp(-|\omega_{\text{yaw}} - c_{\text{yaw}}|^2/0.25)$ |
| (T3) | Z velocity penalty | $-|v_z|^2$ |
| (T4) | Pitch-roll velocity penalty | $-|\omega_{\text{pitch, roll}}|^2$ |
| (T5) | Pitch-roll position penalty | $-|\theta_{\text{pitch, roll}}|^2$ |
| (T6) | Joint nominal differences penalty | $-|q - q^{\text{nominal}}|^2$ |
| (T7) | Joint limits penalty | $-\bar{\mathbb{1}}(0.9q_{\min} < q < 0.9q_{\max})$ |
| (T8) | Joint accelerations penalty | $-|\ddot{q}|^2$ |
| (T9) | Joint torques penalty | $-|\tau|^2$ |
| (T10) | Action rate penalty | $-|\dot{a}|^2$ |
| (T11) | Walking height penalty | $-|h - h_{\text{nominal}}|^2$ |
| (T12) | Collisions penalty | $-n_{\text{collisions}}$ |
| (T13) | Air time penalty | $-\sum_f \mathbb{1}(p_f)(p_f^T - 0.5)$ |
| (T14) | Symmetry penalty | $-\sum_f \bar{\mathbb{1}}(p_f^{\text{left}})\bar{\mathbb{1}}(p_f^{\text{right}})$ |

Table 2: Reward terms that make up the reward function. The coefficients for each term can differ for every robot and are noted in Table 3.

| Robot | T1 | T2 | T3 | T4 | T5 | T6 | T7 | T8 | T9 | T10 | T11 | T12 | T13 | T14 | $T$ |
|---|---|---|---|---|---|---|---|---|---|---|---|---|---|---|---|
| ANYmalB | 2.0 | 1.0 | 2.0 | 0.05 | 0.2 | 0.0 | 10.0 | 2.5e-7 | 2e-4 | 0.01 | 30.0 | 1.0 | 0.1 | 0.5 | 20e6 |
| ANYmalC | 2.0 | 1.0 | 2.0 | 0.05 | 0.2 | 0.0 | 10.0 | 2.5e-7 | 2e-4 | 0.01 | 30.0 | 1.0 | 0.1 | 0.5 | 20e6 |
| Barkour v0 | 3.0 | 1.5 | 2.0 | 0.05 | 0.2 | 0.0 | 10.0 | 2.5e-7 | 2e-4 | 0.01 | 30.0 | 1.0 | 0.1 | 0.5 | 15e6 |
| Barkour vB | 2.0 | 1.0 | 2.0 | 0.05 | 0.2 | 0.0 | 10.0 | 2.5e-7 | 2e-4 | 0.01 | 30.0 | 1.0 | 0.1 | 0.5 | 15e6 |
| Silver Badger | 2.0 | 1.0 | 2.0 | 0.05 | 0.2 | 0.0 | 10.0 | 2.5e-7 | 2e-4 | 0.01 | 30.0 | 1.0 | 0.1 | 0.5 | 12e6 |
| Bittle | 5.0 | 2.5 | 2.0 | 0.05 | 0.2 | 0.0 | 10.0 | 2.5e-7 | 2e-4 | 0.01 | 30.0 | 1.0 | 0.1 | 0.5 | 40e6 |
| A1 | 2.0 | 1.0 | 2.0 | 0.05 | 0.2 | 0.0 | 10.0 | 2.5e-7 | 2e-4 | 0.01 | 30.0 | 1.0 | 0.1 | 0.5 | 12e6 |
| Go1 | 2.0 | 1.0 | 2.0 | 0.05 | 0.2 | 0.0 | 10.0 | 2.5e-7 | 2e-4 | 0.01 | 30.0 | 1.0 | 0.1 | 0.5 | 12e6 |
| Go2 | 2.0 | 1.0 | 2.0 | 0.05 | 0.2 | 0.0 | 10.0 | 2.5e-7 | 2e-4 | 0.01 | 30.0 | 1.0 | 0.1 | 0.5 | 12e6 |
| Cassie | 3.0 | 1.5 | 2.0 | 0.05 | 0.2 | 0.0 | 10.0 | 2.5e-7 | 2e-5 | 0.01 | 30.0 | 1.0 | 0.1 | 0.5 | 50e6 |
| Talos | 4.0 | 2.0 | 2.0 | 0.05 | 0.2 | 0.2 | 10.0 | 2.5e-7 | 2e-5 | 0.01 | 30.0 | 1.0 | 0.1 | 0.5 | 80e6 |
| OP3 | 4.0 | 2.0 | 2.0 | 0.1 | 0.2 | 0.4 | 10.0 | 1.2e-6 | 4e-4 | 6e-3 | 30.0 | 1.0 | 0.1 | 0.5 | 40e6 |
| Nao V5 | 4.0 | 2.0 | 2.0 | 0.1 | 0.2 | 0.15 | 10.0 | 1.2e-6 | 4e-4 | 6e-3 | 30.0 | 1.0 | 0.1 | 0.5 | 40e6 |
| G1 | 3.0 | 1.5 | 2.0 | 0.05 | 0.2 | 0.2 | 10.0 | 2.5e-7 | 5e-5 | 0.01 | 30.0 | 1.0 | 0.1 | 0.5 | 50e6 |
| H1 | 2.0 | 1.0 | 2.0 | 0.05 | 0.2 | 0.2 | 10.0 | 2.5e-7 | 2e-5 | 0.01 | 30.0 | 1.0 | 0.1 | 0.5 | 50e6 |
| Hexapod | 4.0 | 2.0 | 2.0 | 0.05 | 0.2 | 0.0 | 10.0 | 2.5e-7 | 2e-4 | 0.01 | 30.0 | 1.0 | 0.1 | 0.5 | 15e6 |

Table 3: Reward coefficients $r_c$ for all the reward terms for all robots. The last column $T$ is the time-dependent reward curriculum length until all penalties are linearly increased from zero to their final values.

**Domain Randomization.** As known from previous work, the extensive application of Domain Randomization is necessary to ensure that learned policies can be transferred to the real world [1, 13, 46]. Therefore, we randomize as many aspects as possible to learn a robust policy for all robots in the training set and also to ensure that the policy can generalize well to otherwise unseen parameters during zero-shot transfer to new robots. We randomize the robot attributes (trunk mass and inertia, center of mass displacement, feet sizes, joint torque limits, joint velocity limits, joint dampings, rotor inertias, joint stiffnesses, joint friction coefficients and joint control ranges.), the terrain (ground friction coefficient, gravity, contact stiffness and damping), the control (PD gains, action scaling factor, motor strength, joint offsets and actuator delays) and the initial state of the robot (linear and angular velocity, orientation, joint positions and velocities). A new set of these variables is sampled twice per episode on average. Also, to make the policy robust to noisy sensors and learn to act even when some observations are missing, we add observation noise at every time step and randomly drop observations, i.e. set them to zero. Finally, to make the policy robust to external disturbances, we randomly perturb the linear velocity of the robots.

## B    Hyperparameters

The hyperparameters used for training the policies in our experiments are noted in Table 4. We use the same hyperparameters for all architectures to ensure a fair comparison. For the fine-tuning experiments, we reduce the learning rate by $1/3$ to make the learning process more stable and avoid forgetting in the policy by initial overfitting to the new robot.

| Hyperparameter | Value |
|---|---|
| Batch size | 522240 (16 robots * 3 envs * 10880 steps) |
| Mini-batch size | 32640 (16 robots * 2040 samples) |
| Nr. epochs | 10 |
| Initial and final learning rate | 0.0004, 0.0 (Annealed over 100M steps) |
| Entropy coefficient | 0.0 |
| Discount factor | 0.99 |
| GAE $\lambda$ | 0.9 |
| Clip range | 0.1 |
| Max gradient norm | 5.0 |
| Initial action standard deviation | 1.0 |
| Clip range action standard deviation | 1e-8, 2.0 |
| Clip range action mean | -10.0, 10.0 (Before applying $\sigma_a$) |
| Nr. policy network parameters | 430k (+/- 10k for all architectures) |
| (URMA-specific) Initial softmax temperature | 1.0 |
| (URMA-specific) Minimum softmax temperature | 0.015 |

Table 4: The PPO hyperparameters that are used in all experiments for training.

## C    Training Time and Hardware

All multi-embodiment learning experiments are conducted on AMD Ryzen 9 or AMD EPYC CPUs and NVIDIA RTX 3090 or NVIDIA A5000 GPUs. For a single training run with all 16 robots and the hyperparameters noted in Table 4, we allocate 20 CPU cores, 32GB of RAM and 1 GPU. With this setup, training the policy for 100 million steps per robot (1.6 billion steps in total) takes around 2.5 days. The CPU-based MuJoCo simulation with the small amount of parallel environments (48) poses the bottleneck in our training process. Moving to a more powerful GPU-based simulator with thousands of parallel environments would significantly speed up the training.

## D  Ablation on LayerNorm

To evaluate the effectiveness of the LayerNorm layers in the URMA architecture, we perform an ablation study where we remove all LayerNorm layers from the architecture.

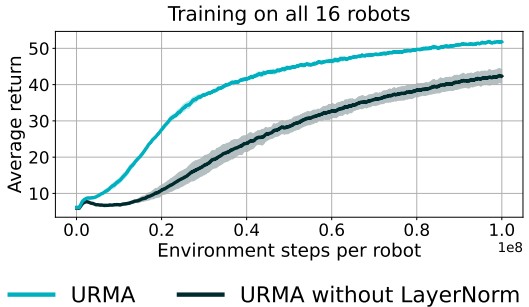

Figure 4: Ablation on the LayerNorm layers in the URMA architecture.

Figure 4 shows that removing the LayerNorm layers leads to a significantly slower learning process and lower final performance.

## E  Ablation on Two Joint Description Encoders

In the URMA architecture, we use two separate encoders for encoding the joint description vectors $d_j$ for the joint observation encoding and the universal action decoding. In the joint encoder, we use the $f_\phi$ MLP and in the universal decoder, we use the $g_\omega$ MLP. To ablate the necessity of having two separate encoders, we replace $g_\omega$ with $f_\phi$ in the universal decoder. None of the gradients are stopped, so $f_\phi$ is optimized for both the joint encoder and the universal decoder. We test the two cases where the representation used for the universal decoder is either taken after the softmax operation ($g_\omega$ = full $f_\phi$) or before the softmax, i.e. after the $\tanh$ operation ($g_\omega$ = partial $f_\phi$).

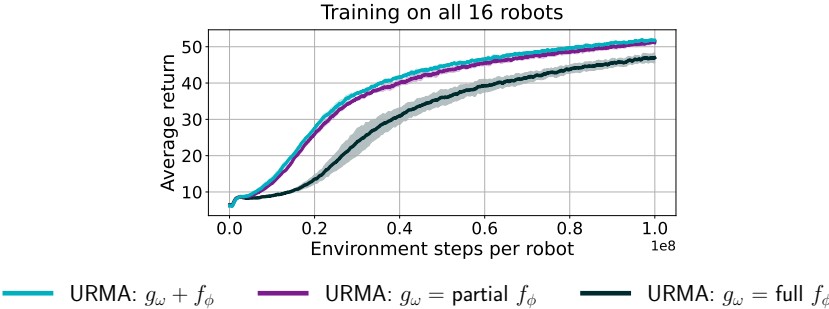

Figure 5: Ablation on the usage of two separate or one single joint description encoder in the joint encoder and the universal decoder.

Figure 5 shows that using the same joint description encoder in both places leads to a slightly worse performance when using the full $f_\phi$ representation and the same performance when using the partial $f_\phi$ representation. Therefore, having two separate encoders for the joint descriptions does not seem to be necessary. However, it might add more flexibility, allowing the policy to learn two sets of encodings when needed.

## F  Ablation on Robot Mass and Dimensions in Observation Space

URMA uses the robot mass $m$ and dimensions $L$, $W$ and $H$ in the general observations $o_g$ and as part of the joint and feet descriptions $d_j$ and $d_f$. To make sure that the policy does not overfit to

specific values of these attributes, we use strong randomization of the mass of the torso and slight randomization of joint nominal positions, which leads to slight changes in the robot bounding boxes (i.e. robot dimensions). To evaluate the necessity of even having the mass and dimensions in the observations, we remove them from the observations and descriptions.

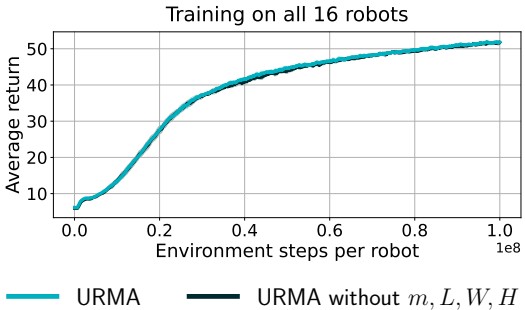

Figure 6: Ablation on the mass and robot dimensions in the observations and descriptions.

Figure 6 shows that removing the mass and dimensions from the observations and descriptions leads to the same performance. Therefore, the policy does not require the robot's mass and dimensions to learn strong locomotion for the different robots.

## G   Ablation on Single Set of Reward Coefficients for All Robots

Getting the most fine-tuned gait for every robot requires tuning the reward coefficients for all robots individually. Mainly, the relative weighting between the two command tracking terms and all the penalty terms needs to be adjusted. More inherently unstable platforms like humanoids need lower penalties at the start of training, to avoid the policy getting stuck in local minima when trying to stabilize and comply with the penalties at the same time. Furthermore, specific joint properties, e.g. joint accelerations or joint torques, need to take on larger values for bigger and heavier robots and also the penalties based on their squared norms (Table 2) scale with the number of joints of the robot. Nethertheless, we only tune 7 out of the 14 reward coefficients for all robots (Table 3), where many of these coefficients are set to the same or similar values for robots of the same morphology. The precise tuning of reward coefficients is not necessary to learn robust policies with good locomotion gaits, rather it is an engineering tool to get fine-grained control over the policy's final behavior.

However, it is possible to use the same set of reward coefficients. To demonstrate this, we perform an ablation study where we train an URMA policy with the same set of reward coefficients for all robots and the same reward curriculum length (URMA: Single reward set) and compare it to a policy with our differently tuned reward coefficients (URMA: Multi reward sets). We evaluate the single reward set policy on all robots using the tuned reward coefficients (URMA: Single reward set, eval multi reward sets) for a fair comparison. We chose the reward coefficients and the reward curriculum length for the single reward set policy to be conservative (Table 5), i.e., we use high tracking coefficients and low penalty coefficients, to ensure that every robot will learn to walk.

| Robot | T1 | T2 | T3 | T4 | T5 | T6 | T7 | T8 | T9 | T10 | T11 | T12 | T13 | T14 | $T$ |
|---|---|---|---|---|---|---|---|---|---|---|---|---|---|---|---|
| All robots | 5.0 | 2.5 | 2.0 | 0.05 | 0.2 | 0.2 | 10.0 | 2.5e-7 | 2e-5 | 6e-3 | 30.0 | 1.0 | 0.1 | 0.5 | 80e6 |

Table 5: Reward coefficients $r_c$ for all the reward terms for all robots. The last column $T$ is the time-dependent reward curriculum length until all penalties are linearly increased from zero to their final values. The description of the reward terms can be found in Table 2.

Figure 7 shows that using the same set of reward coefficients for all robots keeps the stable learning process and its final performance on the differently tuned reward coefficients is only slightly worse

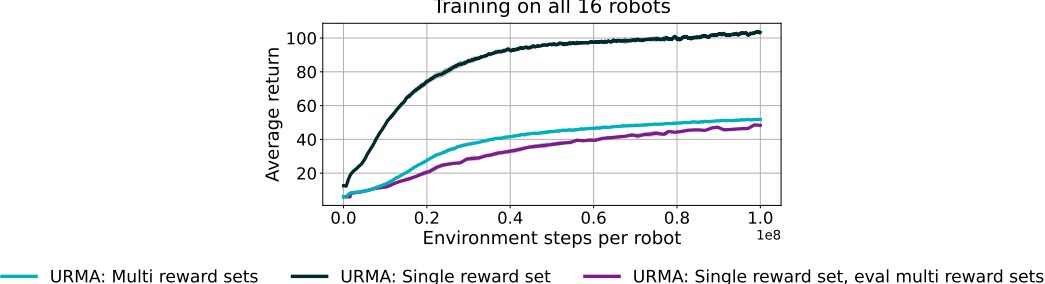

Figure 7: Ablation on using the same set of reward coefficients for all robot embodiments.

than the policy trained directly with the tuned reward coefficients. Therefore, using the same set of reward coefficients for all robots is a valid option to simplify the need for engineering going into the training process and still achieve good locomotion gaits.

## H Ablation on Minimal Quadruped Set for Zero-Shot Transfer

To evaluate the impact of the number of robots and specifically quadrupeds in the training set required for zero-shot transfer to a new quadruped, we perform an ablation study to find a minimal set of quadrupeds that are needed to achieve good zero-shot transfer performance on the Unitree A1. As the Unitree A1 is most similar to the Unitree Go1 and Go2 robots, we train a policy only on those two robots and test if they are already enough to transfer to the A1. Further, we add the Barkour v0 (Bkv0), Barkour vB (Bkvb) and the MAB Robotics Silver Badger (SBg) to the training set to see if the performance improves with these additional robots. Then, we train on all 8 quadrupeds besides the A1, i.e. including the ANYmal B and C and the Petoi Bittle. Finally, we train on the full set of robots besides the A1, so including also the biped, humanoids and hexapod robots.

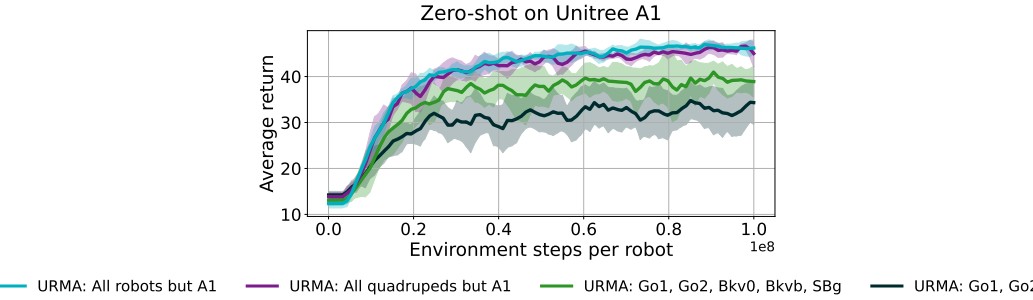

Figure 8: Ablation on the minimal set of quadrupeds needed for zero-shot transfer to the Unitree A1 robot.

Figure 8 shows that training on the Unitree Go1 and Go2 robots already leads to some decent zero-shot transfer performance on the A1. Including the Barkour v0, Barkour vB and the MAB Silver Badger improves the performance further. Training on all quadrupeds besides the A1 leads to the best zero-shot transfer performance and is on par with training on all robots. Therefore, having strongly similar robots in the training set (Go1 and Go2) is enough to get some initial zero-shot transfer performance, but using the most diverse set of quadrupeds leads to the best transferability in the end.

## I Ablation on Humanoids for Zero-Shot Transfer and Fine-Tuning

We evaluate the zero-shot and fine-tuning performance of URMA on the Unitree H1 and the PAL Robotics Talos humanoid robots by following the same procedure as for the MAB Robotics Silver

Badger quadruped (Figure 3 bottom left). We first train the policy on all robots besides the H1 / Talos, while evaluating the zero-shot performance on the H1 / Talos, and then fine-tune the final policy on the H1 / Talos and evaluate the performance again. A standard URMA policy trained on all robots and one trained on only the specific robot from scratch are used as a baseline for comparison.

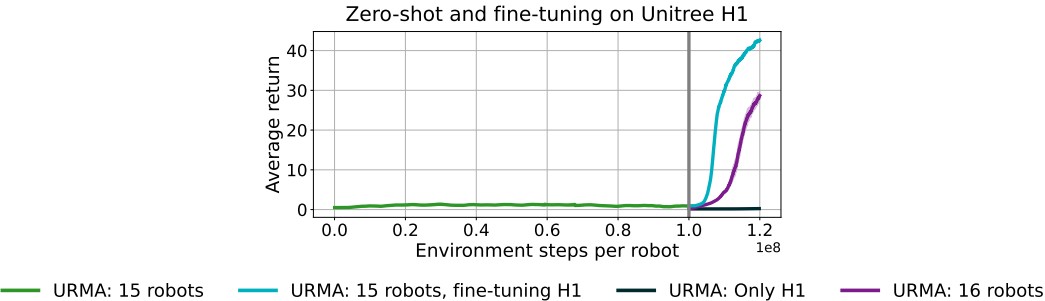

Figure 9: Ablation on the zero-shot and fine-tuning performance of URMA on the Unitree H1 humanoid robot when pre-training on all other 15 robots.

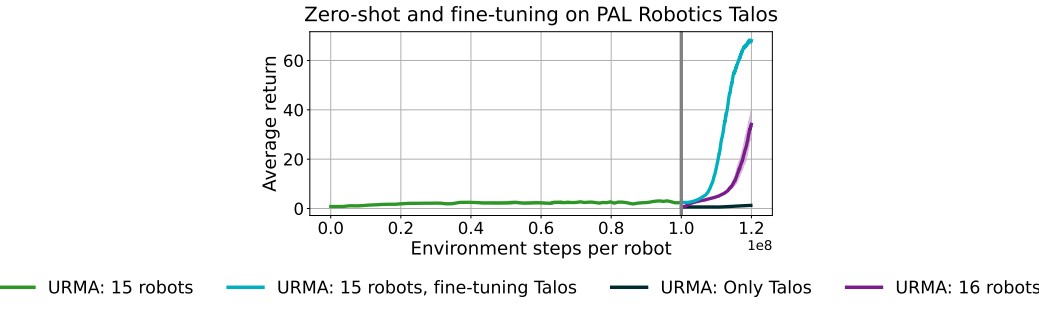

Figure 10: Ablation on the zero-shot and fine-tuning performance of URMA on the PAL Robotics Talos humanoid robot when pre-training on all other 15 robots.

Generalization across our set of biped and humanoid robots is challenging due to the small set of highly variable embodiments and the inherent instability of the humanoid form factor. Figure 9 and Figure 10 show this difficulty, as the policy is not able to do any initial transfer to the Unitree H1 or the Talos. We hypothesize that more biped and humanoid robots in the training set or additional robot randomization on the current robots are necessary for the policy to get some meaningful zero-shot transfer performance. Nethertheless, URMA learns strong general locomotion features during the initial training on other robots, including other humanoids, and is able to quickly adapt to the H1 / Talos during fine-tuning. The fine-tuned policy is able to learn quicker and achieve a higher performance in the 20M steps of fine-tuning than the multi-embodiment and single-robot baseline policies trained from scratch in the same short amount of time. While the fine-tuning policy and the policies trained from scratch will reach the same performance in the end, fine-tuning with an already good representation space allows for quick adaption and fast learning on a new embodiment.

## J Ablation on Inter-Morphology Generalization

Having different robot morphologies in the training set can potentially hurt or help the performance on robots of a specific morphology, leading to inter-morphology generalization. We evaluate the impact of training on additional morphologies by comparing the performance for either quadruped or biped and humanoid robots of a policy trained on all robots with a policy trained only on the respective morphology.

Figure 11 shows that training on additional morphologies does not hurt the performance in the case of quadruped robots and marginally decreases the performance in the case of biped and humanoid

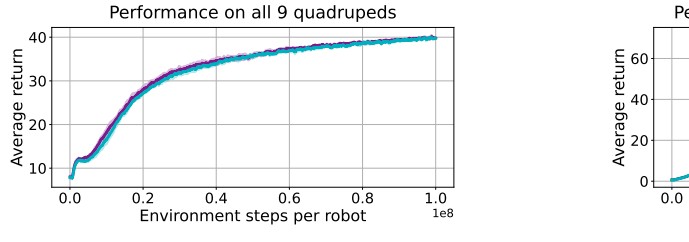

Figure 11: Ablation on the impact of training on additional morphologies when considering only quadruped (Left) or biped and humanoid (Right) robots.

robots. We test the performance on robots that are already in the training set, so it is reasonable that the policy can perform best when training only on these robots. In the quadruped case, the policy seems to have enough capacity to also fully fit the humanoids and hexapod without losing performance. We think that inter-morphology generalization and reasoning about multiple morphologies can still occur and help the policy in the zero-shot and few-shot case, especially when encountering embodiments that mix properties of previously seen robots from different morphologies.

## K  Investigation on Policy Conditioning on Joint Descriptions

The joint description vectors $d_j$ are an integral part in URMA. In the joint encoder, they are used to route the corresponding joint observations to the correct dimension in the joint latent vector $\bar{z}_{\text{joints}}$. In the universal decoder, the joint descriptions are used to unpack the joint specific actions from the action latent vector $\bar{z}_{\text{action}}$. To investigate how strongly the policy conditions on the joint descriptions, we first completely shuffle the joint descriptions for an already trained policy and test it on the Unitree A1 quadruped and the Unitree H1 humanoid. The A1 is able to stand for an all zero velocity command but when receiving any non-zero velocity command, it struggles and falls after one or two steps. The H1 is not able to stand at all and falls mostly immediately, due to the stronger inherent instability of the humanoid morphology. The behavior in both cases does not appear random but rather systematic and strongly uncoordinated between the joints. This makes sense as the policy learned smooth actions but the correct mapping is mixed up.

We also test the conditioning of the policy on different attributes of the joint descriptions. We take the Unitree A1 and strongly perturb the joint descriptions (multiplying the associated values by either $1/3$ or $3$) for the PD gains + action scaling factor, the joints rotor inertias, and the joints friction coefficients. When we increase the values of the description of the PD gains + action scaling factor, the policy assumes the robot's motors and controller are stronger and more responsive than they are. The same situation occurs for the rotor inertia and friction coefficient, the policy assumes that the joints are less responsive and smooth, causing performance degradation.

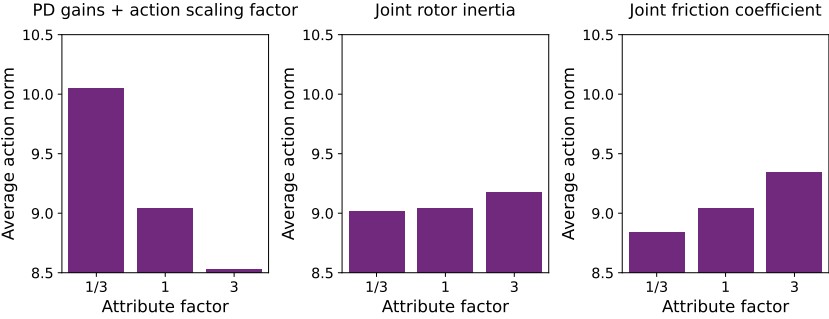

Figure 12: Multiplying the value for attributes of the joint descriptions for the Unitree A1 with different factors. Left – Varying the value for the PD gains + action scaling factor. Middle – Varying the value for the joints rotor inertias. Right – Varying the value for the joints friction coefficients.

The left side of Figure 12 shows that decreasing the description values for the PD gains and action scaling factor by a factor of $1/3$ leads to the policy trying to overcompensate compared to the normal values (factor 1) by producing actions with a higher average magnitude measured over 10 episodes. In the opposite case, increasing the values by a factor of 3 leads to the policy producing actions with a lower average magnitude, as it thinks the robot is more responsive than it actually is. A similar but less pronounced effect can be seen in the middle and right side of Figure 12 for the rotor inertias and friction coefficients. Lower perceived inertias and friction coefficients in the joints (multiplying the values for every joint with $1/3$) leads to the policy producing actions with a lower average magnitude and vice versa. In general, the policy seems to be able to capture the changes in the different attributes of the joint descriptions and adapt its actions to their physical implications.

## L    Individual Robot Returns

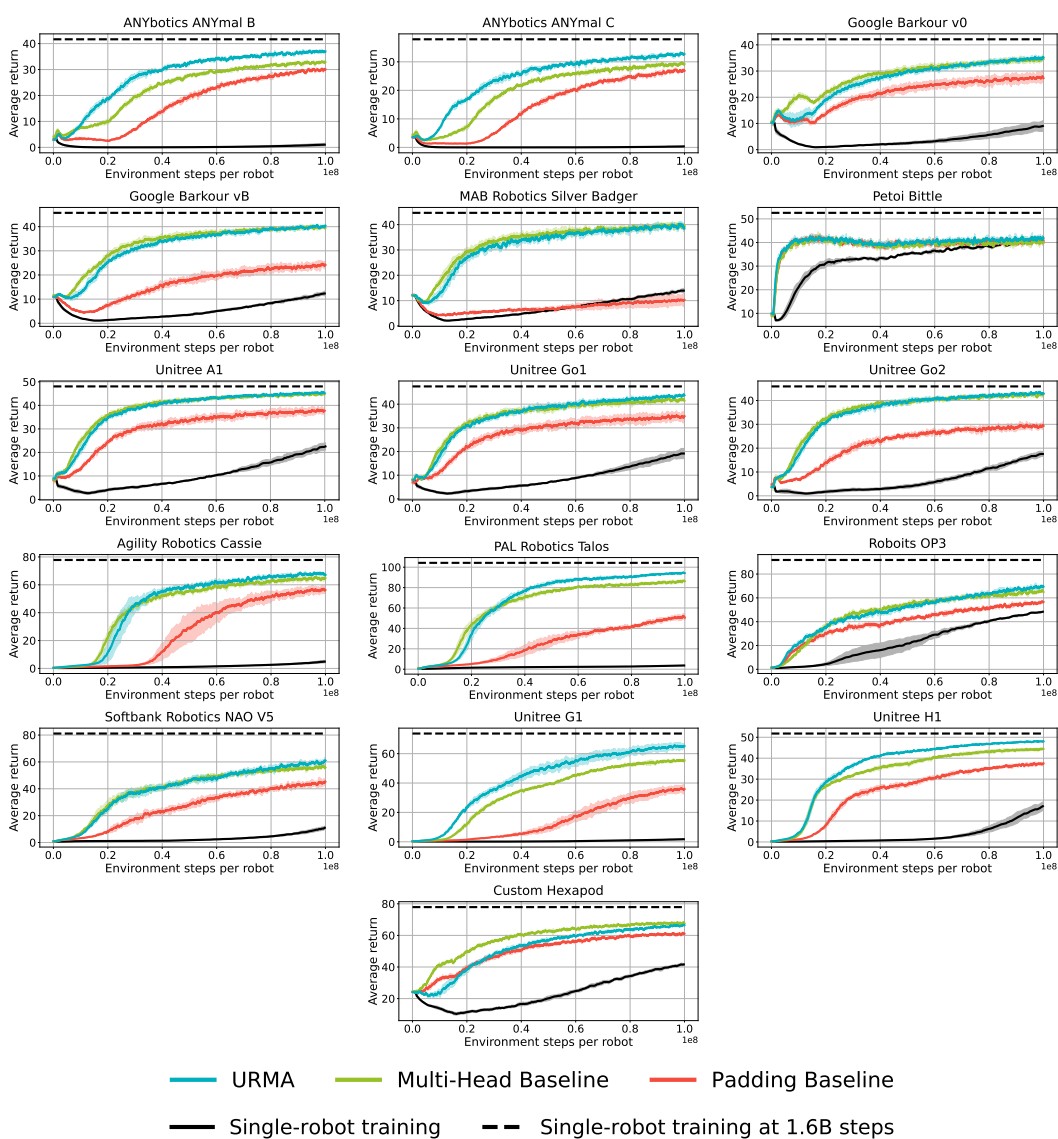

Figure 13: Individual returns for all robots. URMA, the multi-head and the padding baseline are trained on all 16 robots together. The single-robot policy is trained on only one robot. It needs more samples per robot to catch up with the multi-embodiment training. The empirical maximum performance when continuing the training for 1.6B steps on a single robot is shown as a dashed line.

## M  Extended Theoretical Analysis and Proofs

This section provides a detailed proof of Theorem 1.

First, we show that the loss function $\ell$ in (4) is normalized and bounded to $[0, 1]$. The PPO loss—optimized by gradient descent—for a single pair $X_{mi}, Y_{mi}$ is defined as follows:

$$\mathcal{L}(w(h(f(X_{mi}))), Y_{mi}) = -\min\left\{\frac{\pi(a|s,d)}{\pi_0(a|s,d)}A^\pi, \text{clip}(\frac{\pi(a|s,d)}{\pi_0(a|s,d)}, 1-\epsilon, 1+\epsilon)A^\pi\right\}, \qquad (9)$$

where $s$, $d$ and $a$ are the observations, descriptions and actions coming from the sample $X_{mi}$, $\pi(a|s) = w(h(f(X_{mi})))$, $\pi_0(a|s,d)$ is the fixed probability density of the first PPO iteration from $Y_{mi}$ and $A^\pi$ is the true advantage under $\pi$ from $Y_{mi}$. To bound $\mathcal{L}$ we need to bound $A^\pi$ and the ratio $r = \frac{\pi(a|s,d)}{\pi_0(a|s,d)}$ in (9).

By looking at the definition of the advantage $A^\pi = Q^\pi - V^\pi$, we can bound it between $A_{\min} = \frac{R_{\min}}{1-\gamma} - \frac{R_{\max}}{1-\gamma}$ and $A_{\max} = \frac{R_{\max}}{1-\gamma} - \frac{R_{\min}}{1-\gamma}$, where $R_{\min}$ and $R_{\max}$ are the minimum and maximum possible reward. In our training environments, $R_{\min} = 0$ and $R_{\max}$ is the reward given when the policy tracks the given target commands $c$ perfectly. Therefore, in our setting $A_{\min} = -\frac{R_{\max}}{1-\gamma}$ and $A_{\max} = \frac{R_{\max}}{1-\gamma}$.

To bound $r$ in (9), we need to look at the following four cases:

- If $A^\pi > 0$ and $r < 1$ then $r$ is bounded by $1 - \epsilon$.

- If $A^\pi > 0$ and $r > 1$ then $r$ is bounded by $1 + \epsilon$.

- If $A^\pi < 0$ and $r < 1$ then $r$ is bounded by $1 - \epsilon$.

- If $A^\pi < 0$ and $r > 1$ then $r$ is not directly bounded, but we can modify the PPO training to discard $(s, d, a)$ pairs with too big probability ratios, i.e. only allowing samples with $r < 1 + E$ where $E$ is some upper probability ratio that we allow in the case of negative $A^\pi$.

With the bounds on $A^\pi$ and $r$, we can normalize and bound the loss function to be in $[0, 1]$ by defining $\ell$ as

$$\ell(w(h(f(X_{mi}))), Y_{mi}) = \frac{\mathcal{L}(w(h(f(X_{mi}))), Y_{mi}) - l_{\min}}{l_{\max} - l_{\min}}, \qquad (10)$$

where $l_{\min} = -A_{\max}(1+\epsilon) = -\frac{R_{\max}}{1-\gamma}(1+\epsilon)$ and $l_{\max} = -A_{\min}(1+E) = \frac{R_{\max}}{1-\gamma}(1+E)$.

Because our encoder $f$ and decoder $w$ functions take sequences of joint and feet observations and descriptions as input, we rely on the attention mechanism in the encoder and our universal morphology decoder to flatten their respective outputs for the following functions. In practice, in the encoder, we sum over $z_j$ to get rid of the sequence dimension (see (2)), and at the end of the decoder, we concatenate the resulting actions for every joint of a robot to get the flat vector $a = [a^1, a^2, \ldots, a^J]$. To ensure every vector is of the same length, we pad them with zeros.

We start the proof of Theorem 1 by bounding the excess risk, i.e. the difference between the risk $\varepsilon_{\text{avg}}(\hat{f}, \hat{h}, \hat{w})$ where $\hat{f}, \hat{h}, \hat{w}$ are the minimizes of the empirical risk, and the minimum task averaged-risk $\varepsilon^*_{\text{avg}}$. Notice that excess risk represents how much we pay in terms of performance w.r.t. the optimal policy update since we are optimizing our policy using only a finite amount of samples. We

decompose the excess risk, as done in D'Eramo et al. [26], as follows:

$$\varepsilon_{\text{avg}}(\hat{f}, \hat{h}, \hat{w}) - \varepsilon_{\text{avg}}^* = \underbrace{\left( \varepsilon_{\text{avg}}(\hat{f}, \hat{h}, \hat{w}) - \frac{1}{nM} \sum_{mi} \ell(\hat{w}(\hat{h}(\hat{f}(X_{mi}))), Y_{mi}) \right)}_{A}$$

$$+ \underbrace{\left( \frac{1}{nM} \sum_{mi} \ell(\hat{w}(\hat{h}(\hat{f}(X_{mi}))), Y_{mi}) - \frac{1}{nM} \sum_{mi} \ell(w^*(h^*(f^*(X_{mi}))), Y_{mi}) \right)}_{B}$$

$$+ \underbrace{\left( \frac{1}{nM} \sum_{mi} \ell(w^*(h^*(f^*(X_{mi}))), Y_{mi}) - \varepsilon_{\text{avg}}^* \right)}_{C} \tag{11}$$

To bound the complete difference, we will bound its three components $A$, $B$ and $C$. First, $B$ is bounded by 0, as $\hat{f}$, $\hat{h}$ and $\hat{w}$ are the true minimizers of the empirical risk and therefore $f^*$, $h^*$ and $w^*$ can not achieve a lower loss on the empirical dataset. Next, $C$ can be bounded with Hoeffding's inequality with probability $1 - \frac{\delta}{2}$ by $\sqrt{\frac{\ln(2/\delta)}{(2nM)}}$, as it contains only $nM$ random variables bounded in the interval $[0, 1]$. Finally, to bound term $A$, we begin with defining the following auxiliary sets, which make the rest of the derivations cleaner:

- $S = \mathcal{L}(\mathcal{W}(\mathcal{H}(\mathcal{F}(\bar{\mathbf{X}}))), \bar{\mathbf{Y}}) = \{(\ell(w(h(f(X_{mi}))), Y_{mi})) : \ell \in \mathcal{L}, w \in \mathcal{W}, h \in \mathcal{H}, f \in \mathcal{F}\} \subseteq \mathbb{R}^{Mn}$

- $S' = \mathcal{W}(\mathcal{H}(\mathcal{F}(\bar{\mathbf{X}}))) = \{(w(h(f(X_{mi})), Y_{mi})) : w \in \mathcal{W}, h \in \mathcal{H}, f \in \mathcal{F}\} \subseteq \mathbb{R}^{Mn}$

- $S'' = \mathcal{H}(\mathcal{F}(\bar{\mathbf{X}})) = \{(h(f(X_{mi})), Y_{mi}) : h \in \mathcal{H}, f \in \mathcal{F}\} \subseteq \mathbb{R}^{Mn} \times \mathcal{X}$

We start to bound $A$ with probability $1 - \delta$ using Theorem 9 from Maurer et al. [41]:

$$\varepsilon_{\text{avg}}(\hat{f}, \hat{h}, \hat{w}) - \frac{1}{nM} \sum_{mi} \ell(\hat{w}(\hat{h}(\hat{f}(X_{mi}))), Y_{mi})$$

$$\leq \sup_{f \in \mathcal{F}, h \in \mathcal{H}, w \in \mathcal{W}} \left( \varepsilon_{\text{avg}}(f, h, w) - \frac{1}{nM} \sum_{mi} \ell(w(h(f(X_{mi}))), Y_{mi}) \right)$$

$$\leq \frac{\sqrt{2\pi}G(S)}{nM} + \sqrt{\frac{9\ln(\frac{2}{\delta})}{2nM}}. \tag{12}$$

With the contraction lemma, Corollary 11 of Maurer et al. [41], we are bounding $G(S) \leq L(\ell)G(S')$. Compared to Maurer et al. [41], we can't guarantee that the PPO loss is 1-Lipschitz, and therefore, we keep the Lipschitz constant of $L(\ell)$. In practice, this is no problem, as it is just a constant that can be reduced by scaling the loss. Furthermore, we bound the resulting Gaussian complexity of $S'$ with Theorem 12 from Maurer et al. [41], given the universal constants $c_1'$ and $c_2'$ as

$$G(S') \leq c_1' L(\mathcal{W})G(S'') + c_2' D(S'')O(\mathcal{W}) + \min_{y \in Y} G(\mathcal{W}(y)), \tag{13}$$

where $L(\mathcal{W})$ is the largest value for the Lipschitz constants in the decoder function space $\mathcal{W}$, and $D(S'')$ is the Euclidean diameter of the set $S''$. We finish the bounding of the Gaussian complexities of the auxiliary sets by using Theorem 12 from Maurer et al. [41] once more. For the universal constants $c_1''$ and $c_2''$, we bound

$$G(S'') \leq c_1'' L(\mathcal{H})G(\mathcal{F}(\bar{\mathbf{X}})) + c_2'' D(\mathcal{F}(\bar{\mathbf{X}}))O(\mathcal{H}) + \min_{p \in P} G(\mathcal{H}(p)). \tag{14}$$

Combining the bounds on $G(S')$ and $G(S'')$, we have

$$
\begin{aligned}
G(S') \leq{} & c_1' L(\mathcal{W}) \left( c_1'' L(\mathcal{H}) G(\mathcal{F}(\bar{\mathbf{X}})) + c_2'' D(\mathcal{F}(\bar{\mathbf{X}})) O(\mathcal{H}) + \min_{p \in P} G(\mathcal{H}(p)) \right) \\
& + c_2' D(S'') O(\mathcal{W}) + \min_{y \in Y} G(\mathcal{W}(y)) \\
={} & c_1' c_1'' L(\mathcal{W}) L(\mathcal{H}) G(\mathcal{F}(\bar{\mathbf{X}})) + c_1' c_2'' L(\mathcal{W}) D(\mathcal{F}(\bar{\mathbf{X}})) O(\mathcal{H}) + c_1' L(\mathcal{W}) \min_{p \in P} G(\mathcal{H}(p)) \\
& + c_2' D(S'') O(\mathcal{W}) + \min_{y \in Y} G(\mathcal{W}(y)).
\end{aligned}
\tag{15}
$$

We continue to bound the Euclidean diameters in (15) with $D(S'') \leq 2 \sup_{h,f} \|h(f(\bar{\mathbf{X}}))\|$ and $D(\mathcal{F}(\bar{\mathbf{X}})) \leq 2 \sup_{f \in \mathcal{F}} \|f(\bar{\mathbf{X}})\|$. To minimize the last term of (15), we define $d_0 = 0$ and $w(x, o, d_0, a) = 0, \forall w \in \mathcal{W}$, where $x$ is the resulting representation coming from $h$ and $o$, $d$ and $a$ are the observations, descriptions and actions coming from the sample $X_{mi}$, resulting in $\min_{y \in Y} G(\mathcal{W}(y)) = G(\mathcal{W}(0)) = 0$. We finish the bounds on Gaussian complexities of the auxiliary sets by combining them:

$$
\begin{aligned}
G(S) \leq{} & L(\ell)(c_1' c_1'' L(\mathcal{W}) L(\mathcal{H}) G(\mathcal{F}(\bar{\mathbf{X}})) + 2 c_1' c_2'' L(\mathcal{W}) \sup_{f \in \mathcal{F}} \|f(\bar{\mathbf{X}})\| O(\mathcal{H}) \\
& + c_1' L(\mathcal{W}) \min_{p \in P} G(\mathcal{H}(p)) + 2 c_2' \sup_{h,f} \|h(f(\bar{\mathbf{X}}))\| O(\mathcal{W})).
\end{aligned}
\tag{16}
$$

We finalize the bound on term $A$ from (11) by substituting (16) in (12):

$$
\begin{aligned}
& \varepsilon_{\text{avg}}(\hat{f}, \hat{h}, \hat{w}) - \frac{1}{nM} \sum_{mi} l(\hat{w}(\hat{h}(\hat{f}(X_{mi}))), Y_{mi}) \\
& \leq \frac{\sqrt{2\pi}}{nM} L(\ell) \Big( c_1' c_1'' L(\mathcal{W}) L(\mathcal{H}) G(\mathcal{F}(\bar{\mathbf{X}})) + 2 c_1' c_2'' L(\mathcal{W}) \sup_{f \in \mathcal{F}} \|f(\bar{\mathbf{X}})\| O(\mathcal{H}) \\
& \quad + c_1' L(\mathcal{W}) \min_{p \in P} G(\mathcal{H}(p)) + 2 c_2' \sup_{h,f} \|h(f(\bar{\mathbf{X}}))\| O(\mathcal{W}) \Big) + \sqrt{\frac{9 \ln(\frac{2}{\delta})}{2nM}} \\
& = c_1 \frac{L(\ell) L(\mathcal{W}) L(\mathcal{H}) G(\mathcal{F}(\bar{\mathbf{X}}))}{nM} + c_2 \frac{L(\ell) L(\mathcal{W}) \sup_f \|f(\bar{\mathbf{X}})\| O(\mathcal{H})}{nM} \\
& \quad + c_3 \frac{L(\ell) L(\mathcal{W}) \min_{p \in P} G(\mathcal{H}(p))}{nM} + c_4 \frac{L(\ell) \sup_{h,f} \|h(f(\bar{\mathbf{X}}))\| O(\mathcal{W})}{nM} + \sqrt{\frac{9 \ln(\frac{2}{\delta})}{2nM}}.
\end{aligned}
$$

To complete the proof, we compute the union bound between the three components $A$, $B$, and $C$ of (11). Remembering that $B$ is always non-positive, the bound for $A$ holds with probability $1 - \delta$, and the bound for $C$ holds with probability $1 - \frac{\delta}{2}$, we can assume that the probability of violation of both bounds will be $\delta' = \delta + \frac{\delta}{2} = \frac{3}{2}\delta$. With everything in place, we finally bound the overall excess risk as

$$
\begin{aligned}
\varepsilon_{\text{avg}}(\hat{f}, \hat{h}, \hat{w}) - \varepsilon_{\text{avg}}^* \leq{} & c_1 \frac{L(\ell) L(\mathcal{W}) L(\mathcal{H}) G(\mathcal{F}(\bar{\mathbf{X}}))}{nM} \\
& + c_2 \frac{L(\ell) \sup_f \|f(\bar{\mathbf{X}})\| L(\mathcal{W}) O(\mathcal{H})}{nM} \\
& + c_3 \frac{L(\ell) L(\mathcal{W}) \min_{p \in P} G(\mathcal{H}(p))}{nM} \\
& + c_4 \frac{L(\ell) \sup_{h,f} \|h(f(\bar{\mathbf{X}}))\| O(\mathcal{W})}{nM} \\
& + \sqrt{\frac{8 \ln(\frac{3}{\delta})}{nM}},
\end{aligned}
$$

where we have renamed $\delta'$ as $\delta$.

