# OpenReview forum: "One Policy to Run Them All: an End-to-end Learning Approach to Multi-Embodiment Locomotion"
_robot-learning.org/CoRL/2024/Conference — CoRL 2024_

### Official Review · Reviewer_xWHF · 2024-06-29

**Originality:** 3
**Technical Quality:** 3
**Clarity Of Presentation:** 3
**Potential Impact:** 3
**Recommendation:** 2
**Confidence:** 4

**Review:**

This paper demonstrates a system that can learn a unified policy that can control legged robots across different morphologies and sizes.

Strength
Demonstrate that the policy can either transfer zero-shot to a similar robot or can be fine-tuned more efficiently on out-of-distribution robots compared to baseline. The policy also seems to work pretty well on some real quadruped robots.

Weakness
Overall, it seems like a complicated system. How much tuning is needed in the reward function for a new robot to work is unclear. It seems like different robot needs different reward coefficients, so the benefit of a unified policy is kind of limited. Since Go1 and A1 are pretty similar, it is not surprising that the policy can transfer to A1, it would be interesting to investigate the use of joint description. If one changes the description by a lot, will the policy actually make use of that information to make dramatic changes to the output?

There is also a lack of ablations on how many robots are needed to train the initial policy to ensure a good zero-shot capability or as a good basis for finetuning.

**Quality Of The Limitations Section:**

3

**Questions For Rebuttal:**

(1) I don't understand the theoretical analysis part of the paper. What property of the URMA is used in the analysis that other baseline methods do not have? It would be great to explicitly state things like: "Our URMA has this property XXX, and using the property, we can derive this bound (without this property the bound would not work). While other architectures lack this property."

(2) The single robot training baseline performs very badly. Most papers in RL for quadruped use single robot training and they can demonstrate pretty good performance. I wonder where is this gap coming from.

(3) What do the motions of the bipedal robots look like? How important is it to include them for training if all I care about is quadruped robots? How about fine-tuning performance on bipedal robots?

(4) For the quadruped robot with a spine, is the policy fine-tuned to utilize the spine in any way? e.g., does it discover a strategy that quadruped with no spine cannot do?

**Robotics Focus:**

4

**Summary Of Paper:**

This paper presents a archetecture to train a unified policy that can control many different legged robots.

**Summary Of Recommendation:**

I think more analysis/evidence for the benefit of using a unified policy is needed.

---

### Official Review · Reviewer_kaLC · 2024-07-18
**Strong step towards general control policies for arbitrary robot configurations.**

**Originality:** 4
**Technical Quality:** 4
**Clarity Of Presentation:** 4
**Potential Impact:** 4
**Recommendation:** 4
**Confidence:** 5

**Review:**

Overall I think the work is strong and is moving towards an exciting and valuable direction. While I am not fully convinced yet that the results shown here can fully generalize to a diverse set of robots and tasks, the method presented is arbitrarily scalable and definitely has the potential to be general, something that prior works do not have. The results do still seem promising and provide a direction to work towards in having a large robot model that others can use to fine tune for their own platform. This is something that can benefit many researchers in speeding up training time for their own works.

## Strengths:

The proposed method has a truly general way of encoding both the joint (state) space and action space of arbitrary robot morphologies/configurations. The method does not rely on any padding or per task/robot networks and thus has the potential to scale arbitrarily, something that I have not seen from previous works. This is the main strength over previous MTRL works. Given more robots to train over, the number of network parameters does not change, allowing for an actual useable framework for a robot “big data” regime where having more data should only help performance.

I can see the impact of this work being big. If there are more refinements, and it can allow for offline RL use, it can allow for lots of sharing and reuse of good performing policies across the field. Individuals can contribute their own robot datasets, which can be used to improve a large robot foundation model, and can then use it and fine-tune for their own specific platform.

The text and method are for the most part clear and understandable. Explanation details are mostly thorough and go into the sufficient greater detail in the appendix.

The number of training experiments run is appreciated, training policies for 16 works is a lot of work. Though I would like to see some additional comparison experiments (see below).

The technical details of the methods and analysis of the algorithm seem to be all correct to my understanding.


## Limitations:

I think the explanation of what the “joint description vector” is exactly should be moved from the Appendix into the main text. These design choices to me seem highly important, since this is basically your choice of robot agnostic abstraction. How you are choosing to describe robots in a generic way is a main part of your contribution. There is also no exact explanation of what the “feet description” vector contains. That is definitely needed as well.

It would also be interesting to see how much performance are you giving up in favor of generality, if any at all, or even in the extreme case the policy seeing different morphologies is helpful. Presumably, if instead of spending 100 million steps on each robot (so 1.6 billion steps total) you spent all 1.6 billion steps specializing for just a single robot you would get higher performance. How much higher performance is this and is the performance actually higher? Or even if given the same data budget is seeing different morphologies more helpful?

It would also be interesting and useful to see some study on how many robots are needed to get good generality. From your results it seems like you can train on 8 different quadrupeds and still zero-shot generalize to a similar quadruped. Can this still happen if you train on less quadrupeds? I guess the zero-shot test to the Silver Badger kind of shows this; in that case since the Silver Badger is a more different kind of quadruped it’s zero-shot performance is worse. But I would be interested in some experiment to see if there is a “minimal set” of quadrupeds you can train over while still having satisfactory generalization performance.

In a similar line, how much does vastly different morphologies (like quadruped vs. biped/humanoid) help learning of other morphologies? If you remove bipeds and humanoids from the training distribution how does that affect the results on the quadrupeds? So if I only care about generalization to/fine-tuning for a more specific class of robots like quadrupeds or humanoids should I still care about training on different classes of robots?

There are also no generalization experiments done for humanoid robots. If you do similar zero-shot and fine-tuning experiments on one of the humanoid robots as you did for the A1 and Silver Badger what would the performance look like? I think this is necessary to have, in order to show the generalization and quick fine tuning that you argue your method has, and is the entire point of your method in the first place.

The current work is also limited to just diverse robot morphologies, not different tasks (which is understandable, asking for robot and task generalization is a lot for a single work). I would be interested to see if this same method of “foundation model” can provide speed up when fine-tuning for different non-locomotion tasks.

Overall I think the main limitation is that not enough generalization and fine tuning experiments are shown to convince me of the main purpose of this method: that it allows quick fine tuning/generalization to new diverse robots. I would like to see more of the evaluation experiments done for other robot morphologies besides quadrupeds.

**Quality Of The Limitations Section:**

3

**Questions For Rebuttal:**

Is there no MLP encoding for the feet observations/descriptions? (There is no MLP blocks there in Figure 2, they go straight into the attention encoding)

Why is there a different joint description encoding used to produce the mean and std dev of the action? Why can’t the same $f_\phi$ encoding be used?

Why do the single-robot training baselines do so poorly on the quadruped scenarios? You mention that this is because of aggressive DR in order to facilitate sim2real transfer, but from the learning curves it looks like they won’t learn at all, not just that they require more samples (the learning curve seems to have plateaued already). Is this the case or if given more samples will the single-robot baselines actually learn? If not then that is very strange since there are many multiple prior works (that you reference, [1-10]) that can solve the quadruped locomotion task.

Can you provide ranges for the DR used, especially since there seems to be some odd choices of things to randomize? Why would feet size and torque and velocity limits need to be randomized, these should all be pretty well known and fixed right? What is the “action scaling factor”, is this literally some scaling of the policy action? What is the point of that, even if the resulting torque is different on hardware, we know absolutely for certain what action we are trying to command right? What is “motor strength”, is this like some torque scaling factor? Would the randomization of the PD gains not have the same effect as this?

**Robotics Focus:**

4

**Summary Of Paper:**

This paper presents a method for learning a general locomotion policy that can be used on multiple diverse robot platforms, varying from quadrupeds to humanoids to hexapods. This abstract controller can then be zero-shot transferred to unseen platforms of similar nature to those seen in training, i.e. a different unseen quadruped robot. By encoding the information of each robot joint individually and then combining them with an attention head into a latent vector, networks of the same size can be used regardless of the number of joints each robot has. The authors then train their system with 16 different robots and show that it can zero-shot transfer to unseen platforms that are similar nature and can be quickly fine-tuned if the morphology differs a lot.

**Summary Of Recommendation:**

A good step towards the possibility of an actual "robot foundation model". Novel control architecture that can scale and handle robots with arbitrary number of joints/actuators. Shows successful zero-shot transfer and fine-tuning capabilities in limited settings with possibility for more generalization.

---

### Official Review · Reviewer_bzcE · 2024-07-21
**Learning Multi-Embodiment Locomotion based on Multi-Task Reinforcement Learning**

**Originality:** 3
**Technical Quality:** 2
**Clarity Of Presentation:** 2
**Potential Impact:** 3
**Recommendation:** 3
**Confidence:** 5

**Review:**

The paper proposes the Unified Robot Morphology Architecture (URMA) to learn a single locomotion policy for robots with different morphologies. This study uses the Multi-Task Reinforcement Learning (MTRL) concept to share knowledge between tasks and learn a common representation space applicable to all tasks. In the proposed method, embodiment-specific observations are encoded and combined into a single joint latent vector to handle the varying sizes of observations for each robot. To determine the final action for the robot, a universal morphology decoder is employed. This decoder takes the general action latent vector, pairs it with the set of encoded specific joint descriptions, and combines it with the single joint latent vectors to produce the mean and standard deviation of the actions for every joint. Overall, the proposed idea is intriguing, and targeting the challenge of cross-embodiment learning is a nice effort. However, the paper's presentation, comparison, and quantification need significant improvement.

***Comments and Questions***

1.  The idea of using Multi-Task Reinforcement Learning for cross-embodiment is nice, as the framework can handle varying sizes of observation and action spaces. The experiment with quadrupeds looks promising.

2.  It is counter-intuitive that shown, in the paper, the specialized policy training performs distinctly worse than the main policy. It is well-established in the literature that specialized policies can perform agile and challenging locomotion tasks effectively. This limitation should be addressed, as maybe the proposed framework is not suitable for specialization. For a fair comparison, the proposed method should be compared with a simple policy using MLP and the current state-of-the-art RL locomotion techniques (that obviously shows impressive performance in the literature).

3. Comprehensive comparison and quantification of learned gaits in terms of energy efficiency, robustness, and agility should be conducted with a single trained policy and a specialized policy for a single robot with state of the art RL schemes.

4. Surprisingly, there is no video of other embodiments' locomotion except quadruped, which is a central point of the paper. This is a critical issue for conference submission, as all submitted papers should be reviewed under fair and equal conditions, and videos should not be submitted after rebuttal. How can readers understand the locomotion behavior of the other embodiments solely from the reward plot?

5. The authors state this is the first step toward a foundation model for locomotion. "This flexible architecture can be seen as a first step in building a foundation model for legged robot locomotion." I believe this is not the first step in building a foundation model, as there are many other recent efforts in this area. For example please see the following reference as a step toward cross-embodiment of locomotion:

- `Luo, Zeren, et al. "MorAL: Learning Morphologically Adaptive Locomotion Controller for Quadrupedal Robots on Challenging Terrains." IEEE Robotics and Automation Letters (2024).`

6. Regarding the total mass of the robot \( m \), its length \( L \), width \( W \), and height \( H \) being part of the observation space, please mention in the limitations that this information is required for each embodiment. This implies that the network implicitly receives a token for each robot. What about adding a robot with different mass and dimensions that the network has not experienced during training? These parts of the observation space would be out of the training distribution and could lead to unexpected actions.

7. What is the difference between the observation of height above ground \( h \) and height \( H \)? Please clarify this point.

8. What are the smallest and largest robots that can be controlled during locomotion in terms of mass and height? Please provide specific measurements.

9. One limitation of the work is that it requires reward weight tuning for each embodiment. Please mention this in the limitations section.

10. How is the computational complexity? How many hours does it take to train the policy for these 16 robots?

11.  The architecture figure is visually appealing but somewhat confusing. Please improve their clarity and fluency.

12.  Since the paper aims to take a step toward a foundation model for robotics, it is suggested to open-source the code to join the forces with different research groups.

**Quality Of The Limitations Section:**

1

**Questions For Rebuttal:**

Please review the questions mentioned in the review section and check the comments.

**Robotics Focus:**

4

**Summary Of Paper:**

The paper proposes the Unified Robot Morphology Architecture (URMA) to learn a single locomotion policy for robots with different morphologies. This study uses the Multi-Task Reinforcement Learning (MTRL) concept to share knowledge between tasks and learn a common representation space applicable to all tasks. In the proposed method, embodiment-specific observations are encoded and combined into a single joint latent vector to handle the varying sizes of observations for each robot. To determine the final action for the robot, a universal morphology decoder is employed. This decoder takes the general action latent vector, pairs it with the set of encoded specific joint descriptions, and combines it with the single joint latent vectors to produce the mean and standard deviation of the actions for every joint. Overall, the proposed idea is intriguing, and targeting the challenge of cross-embodiment learning is a nice effort. However, the paper's presentation, comparison, and quantification need significant improvement.

**Summary Of Recommendation:**

After considering the rebuttal, I have updated my decision to "weak accept."

---

### Author Rebuttal · Authors · 2024-08-08

The rebuttal material contains the updated paper (changes colored in blue), the updated video (with simulation results), the code base and the single_robot_training.pdf for discussion with the reviewers.

**Update - 13.08.24, 02 am (AoE)**
- Minor adjustments to the paper.pdf for reviewer bzcE

**Update - 12.08.24, 05 am (AoE)**
- Final rebuttal version of the paper.pdf

**Update - 09.08.24, 04 am (AoE)**
- Updated the paper to include preliminary plot for Appendix G, added the Talos zero-shot and fine-tuning experiment and improved colors in plots

---

### Decision · Program_Chairs · 2024-09-04

**Decision:**

Accept

**Comment:**

**Post-rebuttal metareview**:

This paper proposes a novel locomotion policy architecture, URMA (Unified Robot Morphology Architecture), which learns a single policy for different types of legged robots. The cross-embodiment capability and the proposed architecture are pretty novel. Most of the reviewers' concerns are addressed in the rebuttal. Adding more real-world results for non-quadrupedal robots will further strengthen the paper.

----------------
**Pre-rebuttal metareview**:

Strengths:
1. The proposed idea to solve cross-embodiment locomotion is novel and interesting.
2. Real-world experiments on quadrupeds are promising.

Weakness:
1. Presentation needs improvement.
2. Need more fair and comprehensive comparison. Why is the specialized policy worse than the main policy?
3. No real-world experiments for non-quadruped robots, which is the main point of the paper.
4. The “first step towards foundation model” argument needs justification.
5. Other issues mentioned by reviewers such as presentation, training setups, lack of ablation, etc.